# Predicting health behavior in response to the coronavirus disease (COVID-19): Worldwide survey results from early March 2020

**David Anaki**[1,2]*, **Jamie Sergay**[3]

**1** Department of Psychology, Bar-Ilan University, Ramat-Gan, Israel, **2** The Leslie and Susan Gonda (Goldschmied) Multidisciplinary Brain Research Center, Bar-Ilan University, Ramat-Gan, Israel, **3** Department of Biomedical Engineering, University of Wisconsin - Madison, Madison, Wisconsin, United States of America

* david.anaki@biu.ac.il

## Abstract

The current pandemic outbreak of the novel COVID-19, which originated from Wuhan in China in late 2019, has eventually spread to six continents with a rising toll of death cases. No vaccine has yet been developed for COVID-19. The compliance of the general public with the advice and regulations of the health authorities and the adoption of effective health behavior regimens are currently the only weapons to effectively cope with the disease. Here we report the results of a worldwide survey (n = 953) conducted between March 2 and March 14, 2020 that sought (a) to identify critical proximal predictors of health behavior relevant to the current situation, (b) to examine their relationships to various demographic characteristics of the population, (c) and to provide a model of health behavior specific to COVID-19. We found that the perceived severity of the disease and susceptibility to it, emotional reactions, and attitudes toward COVID-19 predicted one-third of the preventive behavior variance. Various demographic variables influenced these predictors. Based on the data collected, we constructed, using path analysis, a theoretical model of health behavior. Our results emphasize the need to consider the impact of antecedent variables on actual precautionary behavior and the influence of demographic factors on these antecedent variables. Understanding the complex interplay of these precursors in health behavior will maximize their beneficial role, eliminate maladaptive prevention patterns, and facilitate the eradication of the disease.

**Data Availability Statement:** All relevant data are within the manuscript and its Supporting information files.

## Introduction

The coronavirus (COVID-19), first reported in China in December 2019, has spread globally with over 50 million confirmed cases and approximately 1.2 million dead [1, 2]. Several factors enhance the present severity of COVID-19 and its health implications. First, no vaccine has been developed for COVID-19, and it will take months before an effective and safe serum will be introduced. Second, the reproduction rate of the disease is estimated between 2–3 [3], higher than seasonal influenza, yet lower than SARS. However, the critical contagious period

**Funding:** The author(s) received no specific funding for this work.

**Competing interests:** The authors have declared that no competing interests exist.

of SARS appears at the second week of illness, when symptoms are already identifiable and patients can be easily quarantined. In COVID-19, however, transmission can occur before or without symptoms' appearance.

The rapid worldwide spread of the coronavirus underlines the importance of the general public compliance with the advice and regulations of the health authorities. Adopting effective health behavior regimens is the only weapon that can be used against COVID-19 until the virus's characteristics are fully understood, validly robust antiviral treatments are determined, and proven vaccines are administered.

Health behavior is broadly defined as any activity undertaken to prevent or detect disease or to improve health and well-being [4]. Several theoretical models of health behavior exist, including the longstanding health belief model (HBM) [5], protection motivation theory (PMT) [6], and theory of planned behavior (TPB) [7]. In the present study, we utilized different constructs from these models to construe an integrated questionnaire and examine aspects of health behavior during the initial outbreak of COVID-19 outside China. The first aim of the study was to identify critical proximal determinants of health behavior, relevant to the current pandemic situation, and to examine their relationships to various demographic characteristics of the population. The second objective was to pinpoint, through regression analyses, factors that significantly predict preventive health behaviors. Finally, the study attempted to model a theoretical account of health behavior based on the empirical data collected in the study (Fig 1). The proposed model contains four theoretically-related independent variables: perceptions, feelings, attitudes, and intentions. It also contains one dependent variable—health behavior. This model reflects our hypothesis that perceptions about COVID-19's severity and susceptibility, feelings related to it, attitudes about the disease, and the intentions to comply with advised preventive measures influence the actual behavior both directly and indirectly.

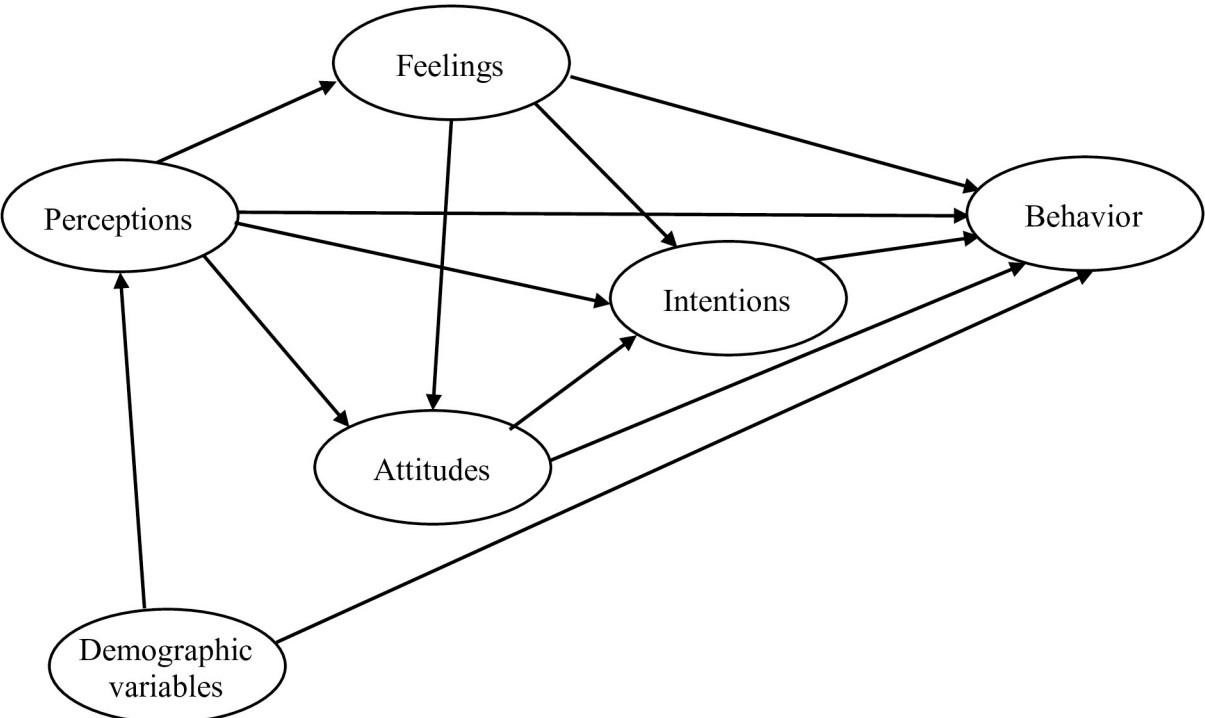

**Fig 1. A theoretical model of health behavior.**

## Materials and methods

### Participants

Participants were recruited from the Amazon's Mechanical Turk (MTurk). MTurk is an online crowdsourcing platform that yields high-quality data comparable with more traditional survey methods [8]. The survey was written using the Qualtrics Survey Software. Before starting the questionnaire participants read a written consent form and approved it by selecting the "Yes" button. The data was collected between March 2 and March 14, 2020. The present research was approved by Bar-Ilan Department of Psychology ethics committee.

A total of 953 participants fully completed the survey. Of the total respondents, 66% were male (Fig 2). Half of the participants were between 25–34 years. The majority of participants were married (51%) and had a Bachelor's degree (53%). 75% of the participants reported being employed full-time, with 38% having income approximately equal to the average national income. Participants came from three continents: North America (67%), Europe (13%), and Asia (20%).

### Questionnaire

The online questionnaire consisted of 25 questions (and nine demographic questions). The survey questions were based on studies that assessed risk perception and precautionary behaviors of the general public in previous pandemic outbreaks (e.g., SARS, avian influenza [9]). Items specific to COVID-19, such as preventive measures, were taken from the website of the Centers for Disease Control and Prevention (www.cdc.gov). Specifically, the questionnaire included the following constructs: perceived severity, perceived vulnerability, feelings of anxiety, attitudes toward the virus, intentions to take preventive measures advised by the health authorities, and actual protective behaviors that are already implemented by the participants. Perceived severity was measured by the following two items "How severe do you think the Coronavirus disease (COVID-19) is?" and "How harmful is the Coronavirus disease (COVID-19) for your health." The responses were provided on a Likert scale from 1-not severe/harmful at all to 5-very severe/harmful. Perceived susceptibility included the items "Do you think that you are susceptible to getting the Coronavirus disease (COVID-19) if you take no preventive measures?" (on a 1–5 scale) and "How likely is it that you will be diagnosed in 2020 with one of the following medical conditions." Participants were asked to refer to several maladies including COVID-19, seasonal influenza, diabetes, heart attack, and HIV. The scale consisted of 3 possible responses (1-unlikely, 2-even, 3-likely). Two items referred to feelings of worry and fear (i.e., "Are you worried/scared about the Coronavirus disease?" on a 1 to 5 scale).

The attitudes towards coronavirus were divided into four categories: proactive, overestimation, passivity and avoidance. The two items that assessed a proactive attitude were "The health authorities in my country should take extra precautionary measures" and "The protective steps taken by my government are too drastic" (reverse-coded). The two items for overestimation were "The threat is exaggerated by the media" and "It will not be as bad as predicted." *Passivity and resignation attitudes were assessed with the items* "We will all be completely powerless," "We just have to accept it," and "There is nothing we can do about the Coronavirus." Finally, avoidance attitudes were examined with two statements "I will stock up and stay indoors" and "I will move to a place without the Coronavirus." For each statement, participants responded whether they disagree (1), were neutral (2), or agree (3).

Participants were asked about their intentions to comply with six precautionary measures recommended by health authorities. These measures included "Wash your hands frequently (with an alcohol-based hand rub, or with soap and water," "Maintain at least 1 meter (3 feet)

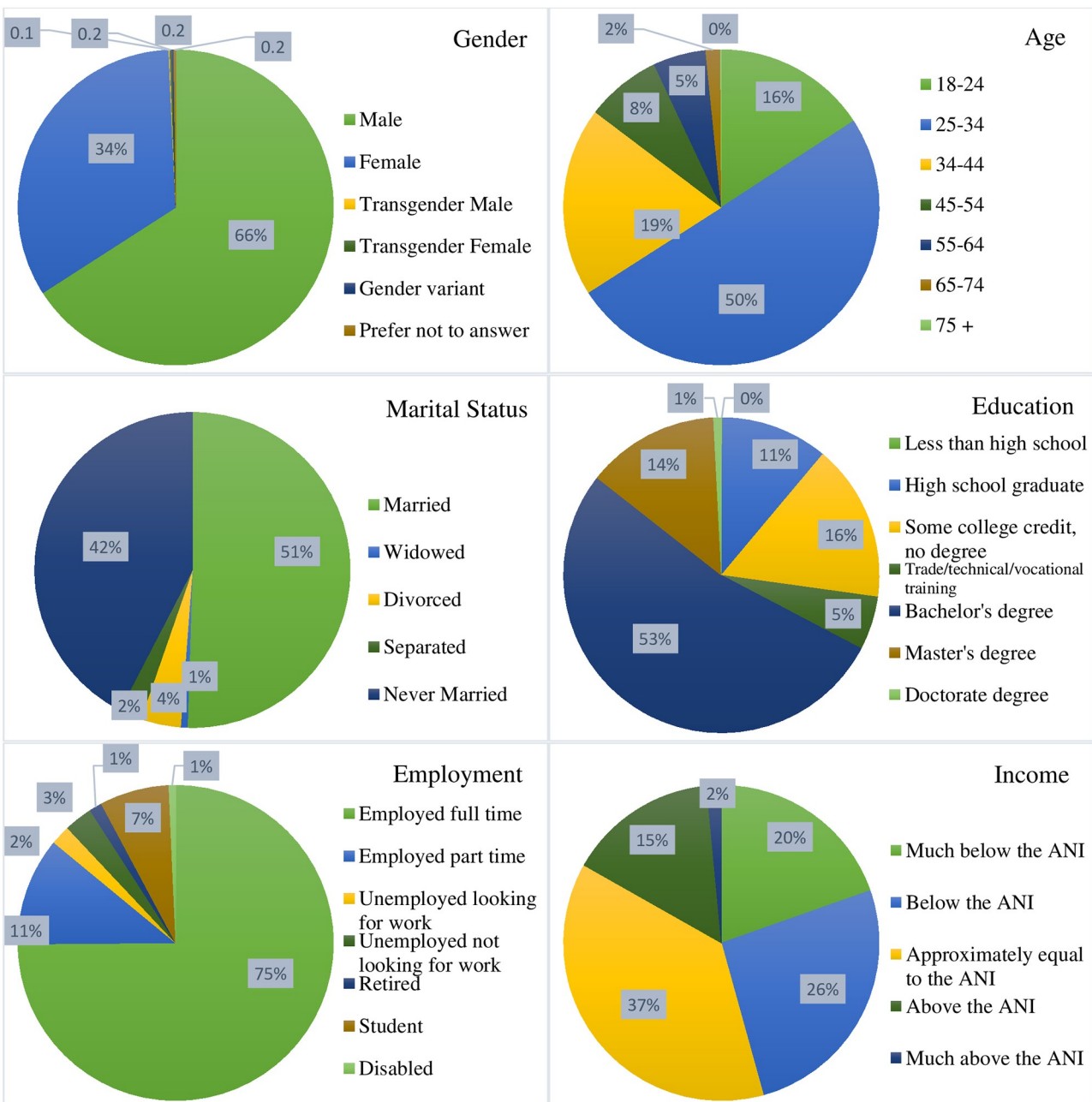

**Fig 2. Demographic distribution of survey participants.**

distance between yourself and anyone who is coughing or sneezing," "Avoid touching your eyes, nose and mouth," "Wear face masks if you are taking care of a person with suspected Coronavirus," "Avoid regions/persons with the Coronavirus," and "Seek medical advice with the onset of symptoms." Responses were provided on a Likert scale from 1-definitely will not take (the advice), to 5-definitely will take (the advice).

Actual protective behavior was evaluated with the following six measures: "I avoid crowded places," "I maintain distance between myself and people who are coughing and sneezing," "I

bought a mouth mask," "I sought medical consultation," "I avoid regions/persons with the Coronavirus," and "I wash my hands frequently." An item indicating lack of preventive behavior ("nothing") was added as well. Participants were allowed to select more than one measure. The survey included two additional short questionnaires that are not relevant to the present study. The three questionnaires were presented randomly to the participants. The demographic questions appeared at the end. The complete questionnaire is available in S1 File.

## Statistical analysis

Chi-square, *t*-tests, and ANOVAS were used to test the statistical significance of differences in demographic groups (e.g., age) regarding the different constructs related to health behavior (e.g., perceived severity). Some variables were dichotomized (e.g., marital status [married, single] or reorganized into fewer levels [e.g, employment [fully-employed, partly-employed, unemployed]]). To avoid false positives resulting from multiple comparisons the alpha Type-I error was set >0.0005.

A multivariate logistic regression analysis was performed to identify factors significantly related with intentions to comply with health authorities' advice and with actual health behavior. A five-step hierarchical regression analysis was conducted. In each step, a group of predictors were entered, and their contribution to the total variance of the criteria, namely behavior, were examined. In the first step of the analysis, we entered demographic variables such as participants' age, gender, marital and employment status, education level, income, date of survey, and continent of origin. Dummy variables were created to incorporate categorical variables (e.g., continent of origin) into the regression analyses. In the second step, we added the variables related to perceived severity and susceptibility to the disease. In the third step, we added two measures of anxious feelings (i.e., fears and worries). In the fourth step, the four attitudes groups (i.e., overestimation, passivity, proactive, and avoidance) were inserted as predictors. Finally, in the fifth step intention to comply with health authorities' advice was entered. An a-priori power analysis (G*Power, [10]) for linear multiple regression, fixed model, $R^2$ increase with an alpha of 0.05, power of 0.80, and 20 predictors revealed that the sample size is sufficient to detect a small-size effect ($f^2$ = .02).

Partial least squares structural equation modeling (PLS-SEM) was used to inspect the theoretical health behavior model presented in the study. PLS-SEM is based on maximizing the explained variance of endogenous latent variables, and it is especially appropriate for exploratory and predictive studies [11]. The analysis was performed with SmartPLS 3.2.7 software [12]. A bootstrap procedure consisting of 10,000 samples was used to test the significance of the coefficients. Power calculation determined that the sample size is large enough to detect a small-size effect (0.80 power, alpha of.05 and 6 predictors, see [13]).

## Results

### Descriptive analysis

The findings reflect the coronavirus-related public health behavior at the beginning of March 2020. Most participants evaluated the severity of COVID-19 as severe or very severe (Fig 3). The mean overall average was 3.76 (/5, SD = 0.96) for severity and 3.65 (SD = 1.08) for harmfulness. Fifty percent of participants responded being quite or very susceptible to COVID-19 ($\bar{x}$ = 3.41, SD = 1.07), with 17% claiming a high likelihood of being diagnosed with the disease. Approximately half of the participants expressed feelings of worry and concern regarding the coronavirus ($\bar{x}$ = 3.44, SD = 1.07, and $\bar{x}$ = 3.24, SD = 1.16, for worry and fear, respectively). The majority of participants endorsed proactive measures. In contrast, overestimation attitudes were also apparent, though passive attitude statements received less support. Medium

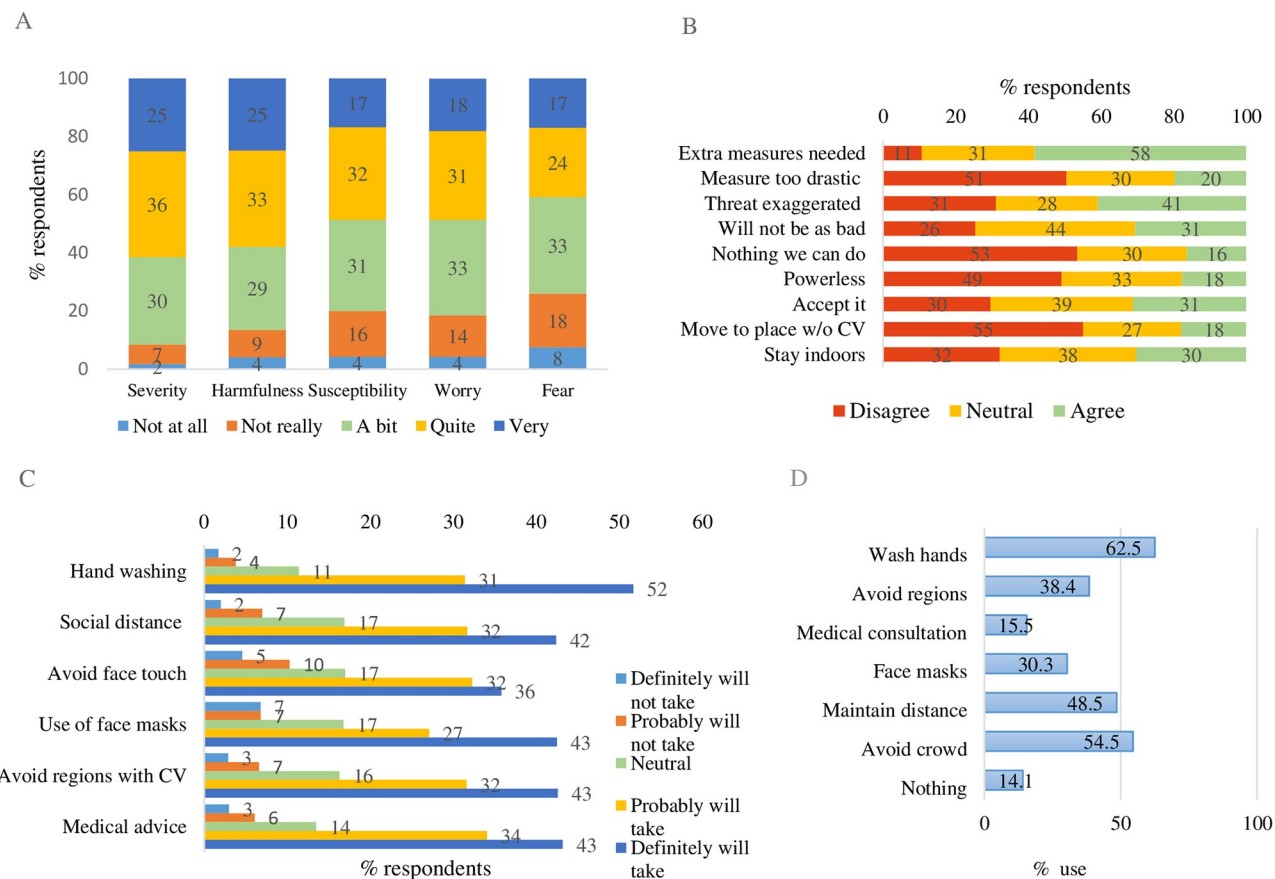

**Fig 3. Survey results regarding (A) perceived severity, susceptibility, and feelings (B) attitudes, (C) intention to comply, and (D) protective behavior.**

support also was obtained for avoidant attitudes. Regarding intentions to comply with health regulation advise the bulk of the participants indicated that they would probably or definitely adopt the advice to wash hands frequently ($\bar{x} = 4.28$, SD = 0.93), keep secure distance from people who cough or sneeze ($\bar{x} = 4.05$, SD = 1.03), avoid touching the face ($\bar{x} = 3.84$, SD = 1.15), wear face masks ($\bar{x} = 3.92$, SD = 1.21), avoid regions with COVID-19 ($\bar{x} = 4.04$, SD = 1.06), and seek medical advice with the onset of symptoms ($\bar{x} = 4.08$, SD = 1.04). The absolute majority of participants were also taking protective measures against the disease, yet 14% reported doing nothing. On average, participants were taking 2.53 (/6, SD = 1.61) protective measures, with 10% adopting more than four. The most common action was hand washing (N = 596), followed by avoiding crowded places (N = 519) and distancing from coughing and sneezing people (N = 462, Fig 3).

## Demographic effects on health behavior and antecedent constructs

Demographic variables influenced all of the health behavior constructs that were examined in the study. Older and married adults perceived coronavirus as more severe and distressing. They perceived themselves as more susceptible to it than young adults. The perceived severity, perceived susceptibility, and feeling of distress of COVID-19 were more pronounced by highly educated and employed participants than by unemployed participants with low levels of education (Table 1).

**Table 1. Perceptions and feelings regarding COVID-19.**

| | Gender (Mean) (M = Male, F = Female) | | | Age (years) (Mean) | | | | | Marital status (Mean) (M = married, S = single) | | | Education (Mean) | | | | |
|---|---|---|---|---|---|---|---|---|---|---|---|---|---|---|---|---|
| **Perceived Severity** | M | F | P | 18–24 | 25–34 | 35–44 | 45+ | p | M | S | p | 0–12 years | 13–15 years | BA | MA+ | p |
| Severity | 3.72 (0.97) | 3.84 (0.94) | < .08 | 3.70 (0.86) | 3.82 (0.98) | 3.69 (0.99) | 3.71 (0.96) | < .24 | 3.81 (1.02) | 3.69 (0.91) | < .06 | 3.64 (0.96) | 3.56 (0.85) | 3.85 (0.97) | 3.84 (1.04) | < .001 |
| Harmfulness | 3.59 (1.10) | 3.77 (1.01) | < .01 | 3.15 (1.25) | 3.76 (1.02) | 3.74 (1.04) | 3.68 (0.96) | < .0005 | 3.81 (1.05) | 3.48 (1.07) | < .0005 | 3.46 (1.05) | 3.17 (1.05) | 3.80 (1.05) | 3.97 (0.97) | < .0005 |
| **Perceived Exposure** | | | | | | | | | | | | | | | | |
| Susceptibility | 3.38 (1.08) | 3.45 (1.06) | < .32 | 3.30 (1.08) | 3.51 (1.08) | 3.36 (1.06) | 3.26 (1.02) | < .03 | 3.55 (1.06) | 3.24 (1.07) | < .0005 | 3.23 (1.10) | 3.09 (1.04) | 3.50 (1.05) | 3.69 (1.07) | < .0005 |
| Diagnosis Like. | 1.77 (0.74) | 1.70 (0.70) | < .16 | 1.93 (0.71) | 1.74 (0.74) | 1.66 (0.69) | 1.66 (0.70) | < .003 | 1.76 (0.76) | 1.74 (0.70) | < .78 | 1.70 (0.62) | 1.69 (0.68) | 1.74 (0.74) | 1.85 (0.82) | < .23 |
| **Feelings of concern** | | | | | | | | | | | | | | | | |
| Worry | 3.40 (1.10) | 3.52 (1.01) | < .10 | 3.28 (1.02) | 3.58 (1.08) | 3.34 (1.12) | 3.25 (0.95) | < .001 | 3.61 (1.03) | 3.23 (1.09) | < .0005 | 3.08 (1.04) | 2.96 (0.94) | 3.61 (1.06) | 3.80 (1.02) | < .0005 |
| Fear | 3.20 (1.17) | 3.34 (1.13) | < .08 | 3.00 (1.10) | 3.42 (1.19) | 3.08 (1.17) | 3.11 (1.01) | < .0005 | 3.49 (1.14) | 2.96 (1.14) | < .0005 | 2.80 (1.05) | 2.69 (0.99) | 3.47 (1.16) | 3.57 (1.09) | < .0005 |
| | Employment (Mean) | | | | Time (Mean) | | | | | Continent (Mean) | | | | | | |
| **Perceived Severity** | Full | Part | Unem | p | Mar. 2–6 | Mar. 7–8 | Mar. 9–11 | Mar. 12–13 | p | Asia | N. America | Europe | p | | | |
| Severity | 3.81 (0.96) | 3.62 (0.98) | 3.59 (0.92) | < .01 | 3.65 (0.99) | 3.72 (0.93) | 3.71 (0.99) | 4.00 (0.91) | < .001 | 4.12 (0.91) | 3.68 (0.96) | 3.61 (0.90) | < .0005 | | | |
| Harmfulness | 3.81 (1.00) | 3.20 (1.13) | 3.16 (1.18) | < .0005 | 3.66 (1.01) | 3.40 (1.21) | 3.70 (0.97) | 3.89 (1.04) | < .0005 | 4.16 (0.92) | 3.67 (0.98) | 2.71 (1.20) | < .0005 | | | |
| **Perceived Exposure** | | | | | | | | | | | | | | | | |
| Susceptibility | 3.48 (1.09) | 3.17 (0.93) | 3.22 (1.04) | < .002 | 3.19 (1.10) | 3.39 (1.05) | 3.42 (1.04) | 3.72 (1.04) | < .0005 | 3.81 (0.97) | 3.31 (1.09) | 3.29 (0.98) | < .0005 | | | |
| Diagnosis Like. | 1.74 (0.75) | 1.73 (0.64) | 1.78 (0.67) | < .83 | 1.56 (0.67) | 1.83 (0.72) | 1.74 (0.75) | 1.87 (0.74) | < .0005 | 1.84 (0.78) | 1.68 (0.71) | 1.93 (0.63) | < .0005 | | | |
| **Feelings of concern** | | | | | | | | | | | | | | | | |
| Worry | 3.57 (1.06) | 3.13 (1.04) | 2.99 (1.00) | < .0005 | 3.16 (1.04) | 3.36 (1.04) | 3.58 (1.08) | 3.72 (1.05) | < .0005 | 3.97 (0.91) | 3.35 (1.08) | 3.06 (0.95) | < .0005 | | | |
| Fear | 3.41 (1.14) | 2.91 (1.03) | 2.66 (1.10) | < .0005 | 3.03 (1.15) | 3.15 (1.14) | 3.34 (1.14) | 3.53 (1.14) | < .0005 | 3.97 (0.98) | 3.13 (1.13) | 2.68 (1.04) | < .0005 | | | |

Unem, Unemployed; Like, Likelihood

Similar effects of age, marital status, employment, and education also affected attitudes towards the disease, albeit not to the same degree (Table 2). Married, employed, and highly educated participants endorsed more avoidant and passive statements. They expressed less proactive attitudes than single, unemployed, and less educated participants. Age, on the other hand, was positively associated with more avoidant and passive attitudes but also with *more* proactive attitudes.

All demographic variables were related to intentions to adopt protective measures (Table 3) and to their actual exercise (Table 4). In general, older, married, highly educated, and employed participants expressed willingness to comply with precautionary measures and actually exercised them more than younger, single, less educated, and unemployed participants.

**Table 2. Attitudes towards COVID-19 across various demographic variables.**

| | Gender (% agree) (M = Male, F = Female | | | Age (years) (% agree) | | | | | Marital status (% agree) (M = married, S = single) | | | Education (% agree) | | | | |
|---|---|---|---|---|---|---|---|---|---|---|---|---|---|---|---|---|
| | M | F | p | 18–24 | 25–34 | 35–44 | 45+ | p | M | S | p | 0–12 years | 13–15 years | BA | MA+ | p |
| *Proactive* | | | | | | | | | | | | | | | | |
| Extra measures needed | 57.5 | 59.7 | < .53 | 44.4 | 54.4 | 73.5 | 66.9 | < .0005 | 54.1 | 61.6 | < .02 | 67.0 | 62.9 | 55.4 | 55.8 | < .07 |
| Measure too drastic | 17.8 | 23.9 | < .02 | 19.2 | 26.4 | 11.4 | 8.6 | < .0005 | 27.4 | 13.2 | < .0005 | 7.5 | 6.8 | 24.0 | 32.6 | < .0005 |
| *Overestimation* | | | | | | | | | | | | | | | | |
| Threat exaggerated | 40.3 | 42.1 | < .62 | 47.0 | 44.1 | 35.1 | 30.2 | < .004 | 43.5 | 37.9 | < .09 | 28.3 | 42.9 | 41.9 | 43.5 | < .05 |
| Will not be as bad | 30.4 | 31.1 | < .44 | 30.5 | 32.0 | 29.2 | 27.3 | < .73 | 31.2 | 30.5 | < .44 | 27.4 | 32.7 | 29.2 | 34.8 | < .46 |
| *Passivity* | | | | | | | | | | | | | | | | |
| Nothing we can do | 15.6 | 18.2 | < .30 | 18.5 | 18.8 | 12.4 | 10.8 | < .05 | 21.4 | 12.7 | < .0005 | 10.4 | 11.7 | 17.1 | 25.4 | < .002 |
| Powerless | 17.0 | 19.5 | < .20 | 11.9 | 24.7 | 9.2 | 11.5 | < .0005 | 22.7 | 12.3 | < .0005 | 10.4 | 7.8 | 20.8 | 26.8 | < .0005 |
| Accept it | 30.1 | 33.0 | < .20 | 25.2 | 31.6 | 35.7 | 30.2 | < .22 | 35.8 | 24.8 | < .0005 | 27.4 | 31.2 | 29.4 | 40.6 | < .07 |
| *Avoidance* | | | | | | | | | | | | | | | | |
| Move to place w/o CV | 18.0 | 17.9 | < .53 | 11.3 | 26.4 | 7.6 | 10.1 | < .0005 | 25.8 | 11.0 | < .0005 | 6.6 | 2.4 | 22.4 | 33.3 | < .0005 |
| Stay indoors | 29.3 | 32.7 | < .16 | 15.9 | 33.3 | 31.9 | 33.8 | < .0005 | 33.5 | 25.7 | < .01 | 25.5 | 25.4 | 31.0 | 39.1 | < .03 |

| | Employment (% agree) | | | | Time (% agree) | | | | | Continent (% agree) | | | | | | |
|---|---|---|---|---|---|---|---|---|---|---|---|---|---|---|---|---|
| | Full | Part | Unem | p | Mar. 2–6 | Mar. 7–8 | Mar. 9–11 | Mar. 12–13 | p | Asia | N. America | Europe | p | | | |
| *Proactive* | | | | | | | | | | | | | | | | |
| Extra measures needed | 58.8 | 61.3 | 53.7 | < .45 | 66.8 | 51.7 | 59.7 | 54.4 | < .003 | 50.8 | 64.0 | 40.3 | < .0005 | | | |
| Measure too drastic | 22.9 | 13.2 | 8.2 | < .0005 | 13.5 | 20.5 | 26.0 | 19.6 | < .007 | 39.4 | 15.9 | 8.4 | < .0005 | | | |
| *Overestimation* | | | | | | | | | | | | | | | | |
| Threat exaggerated | 40.3 | 41.5 | 43.3 | < .78 | 35.9 | 43.6 | 47.2 | 36.3 | < .03 | 48.2 | 36.8 | 50.4 | < .001 | | | |
| Will not be as bad | 31.4 | 29.2 | 26.9 | < .55 | 32.8 | 29.7 | 36.4 | 22.1 | < .01 | 26.4 | 31.8 | 30.3 | < .36 | | | |
| *Passivity* | | | | | | | | | | | | | | | | |
| Nothing we can do | 18.5 | 12.3 | 8.2 | < .006 | 10.8 | 17.4 | 23.4 | 14.2 | < .002 | 26.9 | 14.5 | 9.2 | < .0005 | | | |
| Powerless | 20.9 | 12.3 | 5.2 | < .0005 | 14.3 | 17.8 | 21.2 | 18.1 | < .26 | 35.2 | 15.0 | 4.2 | < .0005 | | | |
| Accept it | 33.1 | 31.1 | 20.9 | < .02 | 29.0 | 29.3 | 30.7 | 36.8 | < .26 | 36.8 | 32.4 | 15.1 | < .0005 | | | |
| *Avoidance* | | | | | | | | | | | | | | | | |
| Move to place w/o CV | 22.2 | 8.5 | 3.0 | < .0005 | 15.1 | 18.9 | 18.2 | 20.1 | < .52 | 43.0 | 12.8 | 5.0 | < .0005 | | | |
| Stay indoors | 33.7 | 24.5 | 17.2 | < .0005 | 23.2 | 26.6 | 32.9 | 41.2 | < .0005 | 39.9 | 31.8 | 6.7 | < .0005 | | | |

Unem, Unemployed

Asian participants perceived the situation as more severe and harmful, and their perceived-susceptibility was higher than North American and European participants. They also expressed more distress than North American, the latter being more worried and fearful than Europeans. North Americans participants expressed more proactive and less over-exaggeration attitudes than Asians and Europeans. Asians endorsed more passive and avoidant

**Table 3. Intentions of adopting protective measures across various demographic variables.**

| | Gender (Mean) (M = Male, F = Female | | | Age (years) (Mean) | | | | | Marital status (Mean) (M = married, S = single) | | | Education (Mean) | | | | |
|---|---|---|---|---|---|---|---|---|---|---|---|---|---|---|---|---|
| | M | F | p | 18–24 | 25–34 | 35–44 | 45+ | p | M | S | p | 0–12 years | 13–15 years | BA | MA+ | p |
| Hand washing | 4.24 (0.95) | 4.34 (0.87) | < .11 | 4.26 (0.75) | 4.17 (0.94) | 4.49 (0.89) | 4.37 (1.06) | < .0005 | 4.20 (0.93) | 4.37 (0.87) | < .003 | 4.37 (0.98) | 4.48 (0.91) | 4.18 (0.93) | 4.27 (0.85) | < .001 |
| Social distance | 3.98 (1.05) | 4.19 (0.96) | < .004 | 3.72 (1.08) | 4.06 (0.99) | 4.19 (1.03) | 4.23 (1.00) | < .0005 | 4.08 (0.99) | 4.05 (1.06) | < .64 | 4.17 (1.12) | 4.04 (1.10) | 4.01 (1.02) | 4.14 (0.84) | < .09 |
| Avoid face touch | 3.73 (1.21) | 4.07 (1.00) | < .0005 | 3.32 (1.32) | 3.91 (1.08) | 4.00 (1.06) | 3.96 (1.18) | < .0005 | 3.95 (1.06) | 3.71 (1.23) | < .001 | 3.78 (1.32) | 3.67 (1.33) | 3.84 (1.08) | 4.16 (0.95) | < .001 |
| Use of face masks | 3.90 (1.22) | 3.96 (1.19) | < .43 | 4.05 (1.17) | 3.93 (1.14) | 3.81 (1.33) | 3.86 (1.35) | < .31 | 3.96 (1.12) | 3.88 (1.28) | < .98 | 4.02 (1.20 | 3.94 (1.34) | 3.85 (1.20) | 4.04 (1.01) | < .32 |
| Avoid regions with CV | 4.03 (1.07) | 4.07 (1.05) | < .56 | 4.05 (1.00) | 3.97 (1.03) | 4.11 (1.14) | 4.19 (1.07) | < .13 | 4.09 (0.97) | 4.02 (1.13) | < .28 | 4.13 (1.14) | 4.15 (1.06) | 3.98 (1.08) | 4.07 (0.88) | < .19 |
| Medical advice | 4.04 (1.06) | 4.18 (1.00) | < .05 | 4.23 (0.96) | 3.97 (1.05) | 4.19 (1.00) | 4.15 (1.10) | < .01 | 4.15 (0.95) | 4.04 (1.11) | < .09 | 4.18 (1.03) | 4.07 (1.18) | 4.02 (1.01) | 4.25 (0.92) | < .11 |
| | Employment (Mean) | | | | Time (Mean) | | | | | Continent (Mean) | | | | | | |
| | Full | Part | Unem | p | Mar. 2–6 | Mar. 7–8 | Mar. 9–11 | Mar. 12–13 | p | Asia | N. America | Europe | p | | | |
| Hand washing | 4.23 (0.95) | 4.37 (0.85) | 4.44 (0.83) | < .03 | 4.34 (0.95) | 4.22 (0.89) | 4.25 (0.92) | 4.29 (0.96) | < .48 | 4.10 (0.88) | 4.32 (0.96) | 4.36 (0.79) | < .01 | | | |
| Social distance | 4.03 (1.09) | 4.06 (0.98) | 4.13 (0.94) | < .54 | 4.18 (0.99) | 3.87 (1.11) | 4.01 (0.99) | 4.17 (0.98) | < .002 | 4.06 (0.92) | 4.15 (1.00) | 3.55 (1.18) | < .0005 | | | |
| Avoid face touch | 4.00 (1.03) | 3.61 (1.28) | 3.21 (1.41) | < .0005 | 3.90 (1.10) | 3.54 (1.27) | 3.96 (1.11) | 4.02 (1.05) | < .0005 | 3.93 (0.97) | 3.98 (1.11) | 2.95 (1.27) | < .0005 | | | |
| Use of face masks | 3.86 (1.21) | 3.97 (1.22) | 4.19 (1.22) | < .01 | 3.89 (1.29) | 3.97 (1.23) | 3.87 (1.17) | 3.93 (1.16) | < .80 | 4.08 (0.87) | 3.80 (1.30) | 4.29 (1.12) | < .0005 | | | |
| Avoid regions with CV | 4.00 (1.06) | 4.13 (1.00) | 4.18 (1.06) | < .14 | 4.09 (1.06) | 3.98 (1.04) | 4.03 (1.08) | 4.07 (1.05) | < .64 | 4.03 (0.94) | 4.03 (1.10) | 4.16 (1.01) | < .44 | | | |
| Medical advice | 4.04 (1.04) | 4.09 (1.11) | 4.32 (0.98) | < .02 | 4.04 (1.07) | 4.16 (1.01) | 4.04 (1.06) | 4.09 (1.01) | < .51 | 4.11 (0.86) | 4.00 (1.11) | 4.47 (0.82) | < .0005 | | | |

Unem, Unemployed

statements than North Americans or Europeans. Asians, North Americans, and Europeans varied in their disposition to comply with various protective measures, but their level of willingness across all measures was similar (x̄ = 4.05, SD = 0.77, x̄ = 4.05, SD = 0.61, x̄ = 3.96, SD = 0.71, respectively, $F(2,950)<1$). Finally, Asians were exercising more precautionary measures ((x̄ = 3.12, SD = 1.53) than North Americans ((x̄ = 2.47, SD = 1.58) or Europeans ((x̄ = 1.90, SD = 1.65, $F(2,950) = 23.741, p < .0001$).

The passage of time naturally affected participants' responses. A gradual increase in severity, susceptibility and emotional distress increased as the disease progressed over time. An increase in agreement with avoidance (i.e., stay indoors) was also observed. Compliance intentions were stable overtime. Time trends were apparent in embracing preventive measures, yet they failed to reach the determined significance.

## Regression analysis

To determine the unique contribution of the different constructs to the exercise of preventive behavior we conducted a five-step hierarchical regression analyses. The first step was significant ($F(9,943) = 8.24, p < .0001, R^2 = .07$) with employment, survey date, and continent of

**Table 4. Protective behavior across various demographic variables.**

| Protective behavior | Gender (% agree) (M = Male, F = Female | | | Age (years) (%% agree) | | | | | Marital status (%% agree) (M = married, S = single) | | | Education (%% agree) | | | | |
|---|---|---|---|---|---|---|---|---|---|---|---|---|---|---|---|---|
| | M | F | $p$ | 18–24 | 25–34 | 35–44 | 45+ | $p$ | M | S | $p$ | 0–12 years | 13–15 years | BA | MA+ | $p$ |
| Nothing | 16.4 | 9.4 | < .004 | 23.2 | 11.3 | 13.5 | 14.4 | < .004 | 10.6 | 18.2 | < .0005 | 18.9 | 18.5 | 12.5 | 9.4 | < .03 |
| Avoid crowd | 54.6 | 54.7 | < .98 | 41.1 | 57.1 | 60.5 | 51.8 | < .001 | 55.5 | 54.2 | < .49 | 59.4 | 51.2 | 55.0 | 53.6 | < .57 |
| Maintain distance | 46.0 | 52.8 | < .05 | 33.1 | 47.9 | 56.2 | 56.8 | < .0005 | 46.8 | 50.2 | < .27 | 55.7 | 49.8 | 46.0 | 50.0 | < .30 |
| Mouth masks | 30.3 | 30.8 | < .86 | 25.2 | 41.0 | 17.8 | 15.8 | < .0005 | 38.0 | 22.1 | < .0005 | 15.1 | 8.3 | 38.7 | 44.2 | < .0005 |
| Medical consultation | 15.1 | 15.4 | < .92 | 11.9 | 22.0 | 7.0 | 8.6 | < .0005 | 20.6 | 9.6 | < .0005 | 3.8 | 2.0 | 19.6 | 29.7 | < .0005 |
| Avoid regions | 42.4 | 30.2 | < .0005 | 39.7 | 39.3 | 37.8 | 34.5 | < .76 | 35.4 | 42.3 | < .02 | 39.6 | 44.9 | 36.9 | 33.3 | < .131 |
| Wash hands | 60.2 | 67.3 | < .03 | 55.0 | 59.0 | 74.1 | 67.6 | < .0005 | 57.1 | 66.4 | < .002 | 68.9 | 70.7 | 60.3 | 53.6 | < .004 |

| Protective behavior | Employment (%% agree) | | | | Time (%% agree) | | | | | Continent (% agree) | | | | | | |
|---|---|---|---|---|---|---|---|---|---|---|---|---|---|---|---|---|
| | Full | Part | Unem | $p$ | Mar. 2–6 | Mar. 7–8 | Mar. 9–11 | Mar. 12–13 | $p$ | Asia | N. America | Europe | $p$ | | | |
| Nothing | 9.2 | 20.7 | 27.7 | < .0005 | 17.0 | 20.8 | 10.8 | 5.4 | < .0005 | 5.7 | 12.9 | 33.6 | < .0005 | | | |
| Avoid crowd | 57.9 | 50.9 | 38.8 | < .0005 | 51.4 | 47.9 | 56.7 | 64.2 | < .003 | 61.1 | 55.9 | 36.1 | < .0005 | | | |
| Maintain distance | 51.3 | 46.2 | 35.1 | < .002 | 45.2 | 46.3 | 45.9 | 58.3 | < .02 | 53.9 | 50.5 | 28.6 | < .0005 | | | |
| Mouth masks | 35.6 | 20.8 | 9.7 | < .0005 | 27.4 | 26.3 | 32.5 | 36.8 | < .06 | 65.3 | 23.7 | 9.2 | < .0005 | | | |
| Medical consultation | 19.1 | 8.5 | 2.2 | < .0005 | 11.6 | 13.9 | 18.6 | 19.1 | < .06 | 34.7 | 11.9 | 4.2 | < .0005 | | | |
| Avoid regions | 37.6 | 40.6 | 41.0 | < .67 | 33.6 | 37.8 | 41.6 | 41.7 | < .22 | 39.9 | 36.2 | 47.9 | < .05 | | | |
| Wash hands | 62.6 | 62.7 | 62.5 | < .99 | 63.3 | 57.1 | 58.4 | 73.0 | < .002 | 55.4 | 65.8 | 56.3 | < .01 | | | |

Unem, Unemployed

origin making significant contributions (Table 5). The second step in the regression yielded a significant change in the accounted variance ($\Delta F(4,939) = 41.26$, $p < .0001$, $\Delta R^2 = .14$). The severity and susceptibility variable made a significant contribution in addition to the variables from the previous step. In the third step, the addition of feelings yielded a positive significant change in the accounted for variance ($\Delta F(2, 937) = 15.05$, $p > .0001$, $\Delta R^2 = .03$). The fourth step was also significant ($\Delta F(4,933) = 10.60$, $p < .0001$, $\Delta R^2 = .03$), due to the fbution of four types of attitudes: proactive, passive, overestimation, and avoidance. In the last step, the intention's variable added a $\Delta R^2 = .06$ to the behavior's variance. In total, the different predictors predicted 33% of the preventive behavior variance.

## PLS-SEM path analysis

In the final part of the analysis we modeled a path model of health behavior using a Partial Least Squares (PLS) SEM analysis [12]. The results of the structural model were examined following the standard guidelines (Fig 4) [13]. First, we assessed the results of the formative measurement model for collinearity between the indicators, namely the relationships between the indicators (e.g., severity) and the latent variables (e.g., perception). We used the variance inflation factor (VIF) value. All the values of VIF were below the threshold of 5, confirming that

**Table 5. (Un)standardized regression coefficients predicting COVID-19 preventive.**

| Variable | Step 1 | | | | Step 2 | | | | Step 3 | | | | Step 4 | | | | Step 5 | | | |
|---|---|---|---|---|---|---|---|---|---|---|---|---|---|---|---|---|---|---|---|---|
| | B | SE | β | p | B | SE | β | p | B | SE | β | p | B | SE | β | p | B | SE | β | P |
| (Constant) | 2.80 | .38 | | < .001 | 0.15 | .41 | | < .71 | 0.09 | .41 | | < .84 | -0.22 | .55 | | < .69 | -1.54 | .54 | | < .004 |
| Gender | 0.03 | .11 | 0.01 | < .78 | 0.10 | .10 | 0.03 | < .31 | 0.12 | .10 | 0.03 | < .24 | 0.06 | .10 | 0.02 | < .55 | 0.14 | .10 | 0.04 | < .15 |
| Age | 0.07 | .05 | 0.05 | < .18 | 0.04 | .05 | 0.03 | < .36 | 0.06 | .05 | 0.04 | < .21 | 0.01 | .05 | 0.01 | < .77 | 0.01 | .05 | 0.01 | < .79 |
| Marital status | 0.00 | .11 | 0.00 | < .98 | -0.05 | .10 | -0.02 | < .61 | -0.10 | .10 | -0.03 | < .31 | -0.03 | .10 | -0.01 | < .77 | -0.08 | .10 | -0.03 | < .39 |
| Education | -0.06 | .05 | -0.05 | < .17 | -0.11 | .04 | -0.08 | < .02 | -0.13 | .04 | -0.10 | < .002 | -0.11 | .04 | -0.09 | < .008 | -0.10 | .04 | -0.07 | < .02 |
| Employment | -0.19 | .09 | -0.09 | < .03 | -0.13 | .08 | -0.06 | < .10 | -0.10 | .08 | -0.04 | < .24 | -0.10 | .08 | -0.05 | < .20 | -0.13 | .08 | -0.06 | < .09 |
| Income | 0.09 | .06 | 0.06 | < .10 | 0.07 | .05 | 0.04 | < .19 | 0.06 | .05 | 0.04 | < .25 | 0.07 | .05 | 0.04 | < .16 | 0.06 | .05 | 0.04 | < .25 |
| Survey date | 0.18 | .05 | 0.12 | < .001 | 0.11 | .04 | 0.08 | < .01 | 0.09 | .04 | 0.06 | < .03 | 0.10 | .04 | 0.07 | < .02 | 0.10 | .04 | 0.07 | < .01 |
| Cont. 1 (Am.) | -0.67 | .14 | -0.20 | < .001 | -0.35 | .13 | -0.10 | < .007 | -0.28 | .13 | -0.08 | < .03 | -0.34 | .13 | -0.10 | < .009 | -0.34 | .13 | -0.10 | < .006 |
| Cont. 2 (Eu.) | -0.90 | .22 | -0.18 | < .001 | -0.47 | .21 | -0.10 | < .03 | -0.42 | .21 | -0.09 | < .04 | -0.34 | .21 | -0.07 | < .11 | -0.39 | .20 | -0.08 | < .06 |
| Susceptibility | | | | | 0.24 | .05 | 0.16 | < .001 | 0.16 | .05 | 0.10 | < .004 | 0.14 | .05 | 0.09 | < .01 | 0.11 | .05 | 0.07 | < .04 |
| Corona diag. like. | | | | | -0.07 | .07 | -0.03 | < .35 | -0.12 | .07 | -0.05 | < .10 | -0.13 | .07 | -0.06 | < .07 | -0.16 | .07 | -0.07 | < .02 |
| Severity | | | | | 0.30 | .06 | 0.18 | < .001 | 0.18 | .07 | 0.11 | < .006 | 0.12 | .07 | 0.07 | < .07 | 0.08 | .06 | 0.05 | < .21 |
| Harmful | | | | | 0.25 | .06 | 0.17 | < .001 | 0.17 | .06 | 0.11 | < .005 | 0.13 | .06 | 0.08 | < .04 | 0.07 | .06 | 0.05 | < .22 |
| Worry | | | | | | | | | 0.24 | .07 | 0.16 | < .001 | 0.20 | .07 | 0.13 | < .004 | 0.16 | .07 | 0.11 | < .02 |
| Fear | | | | | | | | | 0.13 | .07 | 0.09 | < .05 | 0.12 | .06 | 0.09 | < .07 | 0.11 | .06 | 0.08 | < .07 |
| Proactive | | | | | | | | | | | | | 0.36 | .10 | 0.13 | < .001 | 0.14 | .09 | 0.05 | < .13 |
| Over-estimation | | | | | | | | | | | | | -0.15 | .07 | -0.06 | < .05 | -0.15 | .07 | -0.06 | < .04 |
| Passivity | | | | | | | | | | | | | -0.2 | .10 | -0.08 | < .03 | -0.15 | .10 | -0.05 | < .12 |
| Avoidance | | | | | | | | | | | | | 0.43 | .10 | 0.17 | < .001 | 0.36 | .10 | 0.14 | < .001 |
| Intentions | | | | | | | | | | | | | | | | | 0.63 | .07 | 0.28 | < .001 |
| R-squared | .07 | | | | .21 | | | | .24 | | | | .27 | | | | .33 | | | |
| Adjusted R-squared | .06 | | | | .20 | | | | .22 | | | | .25 | | | | .32 | | | |

Cont, Continent; Am, America; Eu, Europe; Diag, Diagnosis; Like, Likelihood.

collinearity was not an issue in the model. Then, the significance and the relevance of the indicators were evaluated by inspecting their outer weights. These were all significant.

Next, the structural model results were evaluated. No collinearity issues were apparent as all VIFs were below 5. Then, the sizes and significance of the path coefficients that reflect the hypotheses were examined. The empirical results showed direct, significant, and positive relationships between perceptions (95% CI: -.26 to -.03), feelings (95% CI: -.26 to -.03), attitudes (95% CI: -.26 to -.03), intentions (95% CI: -.26 to -.03), and preventive behavior. Perceptions were also indirectly related to behavior through feelings ($\beta$ = .12, 95% CI:.06-.18, $p$ < .0001), attitudes ($\beta$ = .07, 95% CI:.04-.11, $p$ < .0001), and intentions ($\beta$ = .09, 95% CI:.06-.12, $p$ < .0001), thus demonstrating partial mediation. In contrast, intentions did not mediate the association between feelings, attitudes, and behavior. Finally, the demographic variables were related to behavior both directly (95% CI: -.13-.02, for both employment and education) and indirectly (e.g., the indirect education-perception-feelings-behavior relationship: $\beta$ = .02, 95% CI:.01-.11, $p$ < .03).

The model's goodness of fit was assessed by measuring the standardized root mean square residual (SRMR) [14]. The SRMR value of 0.075 confirmed the overall model fit. The in-sample predictive power of the model was assessed using the coefficient of determination ($R^2$). The $R^2$ of behavior was equal to 0.31. The out-of-sample predictive value was evaluated using the blindfolding procedure (omission distance = 7). A $Q^2$ value above zero in the cross-validated redundancy report confirms predictive relevance. All $Q^2$ values were significantly

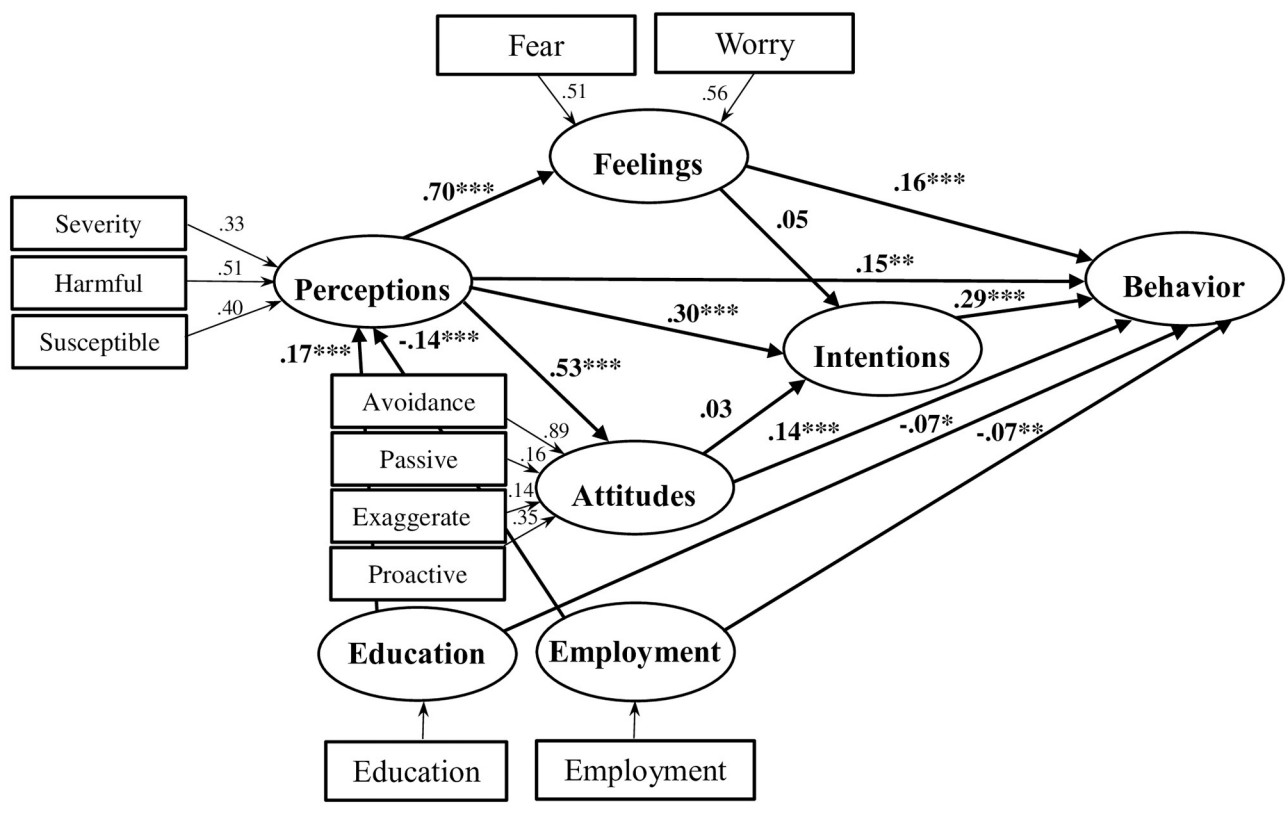

**Fig 4. PLS-SEM path analysis results.**

greater than zero. The effect sizes ($f^2$) of perceptions on feelings and attitudes were large (0.98 and 0.40, respectively). Yet, the direct effects of perceptions and feelings on behavior were small (~0.02). The intentions' effect size on behavior was small-medium (0.11).

## Discussion

In the present study, we investigated several theoretical constructs related to present COVID-19 health behavior and their associations with various demographic variables. We also attempted to identify which of these constructs predict precautionary measures required for preventing infection by the virus. Finally, we modeled a theoretical account of health behavior to delineate causal relationships between the different constituents of the model.

The majority of participants evaluated the coronavirus as severe and harmful and considered themselves as likely to incur it. Participants also expressed feelings of concern regarding the coronavirus. Despite the high percentage of participants (40% to 60%) expressing concern and realistic perception of the disease, these rates were lower than those reported in a study conducted in mainland China in early 2020 [15]. In that study, for example, 85% of participants felt "very" or "extremely" nervous about the disease. This finding may be because the latter study was comprised of Chinese participants who were in the epicenter of the outbreak at the time. Our research, in contrast, consisted of 80% Americans and Europeans. At the beginning of March, these areas were lagging in proper management of the outbreak. The present study provides some evidence for this claim from the finding that the continent of residence was a significant predictor of preventive behavior. Both Americans and Europeans reported adopting less precautionary actions than Asians.

Most participants encouraged the adoption of proactive approach and sanctioned the passive stance towards the disease. Alarmingly, overestimation attitudes were widespread, and a tendency to belittle the situation was apparent in a sizeable portion of the sample (30% to 40%). Self-quarantine, an avoidance attitude that had since become modus operandi in many countries, received moderate support. As time passed, an increase in endorsing this attitude was seen. However, even at its peak (March 12–13), only 40% endorsed the home isolation statement.

Intentions to comply with health authorities' advice was robust, ranging between 68% to 83% for different preventive measures. Yet, actual behavior was much less pronounced. Washing hands, the most popular protective behavior, was exercised by only 62% of participants. Only 10% applied more than four preventive measures. In Zhan et al., (2020) study, 88% of subjects reported using more than four (out of six total) protective measures [15]. To summarize, the picture revealed in our survey indicates partial recognition of the outbreak consequences and limited preparedness to cope with it.

Effects of demographic variables on the different constructs of health behavior examined in the study are ostensible. The relationship of various demographic variables to different precautionary measures emphasizes that the authorities' health behavior campaigns should not be uniform. As recent findings indicate, healthcare providers should implement tailor-made policies to maximize the efficacy of COVID-19 preventive measures, and its maximal propagation to all quarters of the population [16–18].

Some of the insignificant results are also of interest. For example, the lack of difference in perceptions, attitudes, and prevention measures between males and females is disturbing since greater number of men are infected with the virus than women (5% vs. 3%) [19]. Extra preventive behavior is warranted in males, and education efforts should be concentrated in this slice of the population.

Several significant predictors of health behavior were found in the regression analysis, accounting for 33% of the dependent variable variance. Among the demographic variables, education, survey date, and continent of origin were significant predictors of practicing prevention measures. These three demographic variables could be regarded as indirect measures of perceived knowledge about COVID-19. Other predictors were perceived susceptibility and feelings of distress. In addition, attitudes towards COVID-19 prevention also predicted COVID-19 preventive health behaviors. Finally, intentions to comply with health regulations were also correlated with an increase in health behavior.

The current findings mirror similar results that were found in Hong Kong after the SARS outbreak in 2003 [20]. Specifically, they found that perceived susceptibility, self-efficacy (not assessed in the current study), age, and favorite attitudes towards SARS prevention models were significant predictors of preventive measures increase. The predictive role of intentions, perceived severity and susceptibility, and attitudes is compatible with the existing literature that also shows a similar effect. Moreover, a recent study examined the determinants of social distancing in the context of the COVID-19 pandemic from the theory of planned behavior viewpoint [21]. Their findings indicated that intention and habit were significant predictors of social distancing behavior. Moreover, in line with the social cognition model adopted in that study, both subjective and moral norms, and perceived behavioral control were predictors of intention (see also [22], who investigated a similar model to explain COVID-19 preventive behaviors).

The predictive characteristics of feelings, such as worry and fear, that were found here were reported in current studies that addressed the association between fear and COVID-19 preventive behavior [23, 24]. The cumulative set of results, however, highlight a complex relationship between these two constructs. Specifically, in the general population, fear increased preventive

measures [23], but it decreased health behavior among people with mental illness [24]. Social media also influence fear, and the public's problematic information may increase fear and distress, impeding effective coping with the disease [25].

The exploratory path analysis model revealed that perceptions, feelings, attitudes, and intentions have direct paths connections to behavior. Interestingly, partial mediations were also found. Intentions, attitudes, and feelings mediated the relationship between perception and behavior. These findings are partially at odds with many models that view intentions as an intervening variable that mediates between social cognitive variables and behavior [4]. The present findings clearly show that feelings and attitudes are directly related to behavior and that they are themselves mediators between perceptions and behavior. These new findings should be considered in future models of health behavior.

Several limitations exist in the study that ought to be remedied in future research. First, the survey used the Amazon Mechanical Turk platform, a platform based on a non-probability-based convenience sample that does not necessarily represent the population [26]. Future online research should use platforms that allow more diversity in their participants' demographic characteristics [27]. Moreover, we administered the survey in English. Although we used plain language, it may have proved some challenge to non-native English speakers. Finally, cross-national variability is known to influence health behavior [28]. Physical environment, socio-cultural factors, economics, and political settings may be relevant in understanding and predicting health behavior. Future studies should take these environmental settings into account when considering their role in specific nations' health behavior.

A second limitation of the current study lies in its theoretical underpinnings. The integrated model which we present here is based on various theoretical constructs that we collected from several models of health behavior. Other essential components, however, were not included in the study. For example, we did not implement in our model the decisional balance between the relative weight of perceived benefits versus the perceived costs involved in engaging in particular health behavior. Ensuing research examining future outbreaks should systematically investigate existing models in full and compare them to assess their predictive validity and theoretical value. Indeed, several recently published studies have adopted one model through which they examined preventive behavior in the current pandemic context, such as the theory of planned behavior [21] or the health action process approach [29]. Also, since the present survey was launched, several scales were developed that assess both COVID-19 related health behavior [24] and specific predictors such as fear [29], anxiety [30], obsession [31], and stress [32]. Their use in future studies addressing the issue of health behavior is warranted.

On March 2, 2020, when the present survey was launched, there were 80,174 confirmed cases in China and 8,774 cases worldwide. On March 29, there were 82,356 cases in China, a monthly increase of 3%. Worldwide, however, there were 634,835 cases of coronavirus, a staggering increase of 7135% in a month. These numbers reflect a colossal failure of the multilateral health systems, national and international, to contain the disease. One of the missteps in the futile attempt to manage the pandemic was to guarantee that individuals adopt timely and adequate health behavior practices. This dismal health deployment is reflected in the current findings. However, the present study also highlights several essential components that impact the adoption of health measures in the general public. These findings may be implemented in both the current COVID-19 eruption or in the next new or reemerging disease outbreaks.

## Supporting information

**S1 File. COVID-19 questionnaire.**
(DOCX)

**S1 Dataset. Survey raw data.**
(SAV)

## Author Contributions

**Conceptualization:** David Anaki.

**Formal analysis:** David Anaki.

**Methodology:** David Anaki, Jamie Sergay.

**Project administration:** David Anaki.

**Software:** Jamie Sergay.

**Writing – original draft:** David Anaki.

**Writing – review & editing:** Jamie Sergay.

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
