## [Decision Letter · Decision Letter 0]

28 Sep 2020

PONE-D-20-25877

Predicting health behavior in response to the coronavirus disease (COVID-19):

Worldwide survey results from early March 2020

PLOS ONE

Dear Dr. Anaki,

Thank you for submitting your manuscript to PLOS ONE. After careful consideration, we feel that it has merit but does not fully meet PLOS ONE’s publication criteria as it currently stands. Therefore, we invite you to submit a revised version of the manuscript that addresses the points raised during the review process.

We look forward to receiving your revised manuscript.

Kind regards,

Amir H. Pakpour, Ph.D.

Academic Editor

PLOS ONE

Journal Requirements:

2. In the ethics statement in the Methods and online submission information, please specify the type of informed consent that was obtained from the participants (for instance, written or verbal, and if verbal, how it was documented and witnessed).

3. Please provide further details on sample size and power calculations.

4. Please include additional information regarding the survey or questionnaire used in the study and ensure that you have provided sufficient details that others could replicate the analyses. For instance, if you developed a questionnaire as part of this study and it is not under a copyright more restrictive than CC-BY, please include a copy, in both the original language and English, as Supporting Information.

5. In the Methods, please discuss whether and how the questionnaire was validated and/or pre-tested. If this did not occur, please provide the rationale for not doing so.

Reviewers' comments:

Reviewer's Responses to Questions

**Comments to the Author**

1. Is the manuscript technically sound, and do the data support the conclusions?

Reviewer #1: Partly

2. Has the statistical analysis been performed appropriately and rigorously? 

Reviewer #1: Yes

3. Have the authors made all data underlying the findings in their manuscript fully available?

Reviewer #1: Yes

4. Is the manuscript presented in an intelligible fashion and written in standard English?

Reviewer #1: Yes

5. Review Comments to the Author

Reviewer #1: The study entitled “Predicting health behavior in response to the coronavirus disease (COVID-19): Worldwide survey results from early March 2020” examined a meaningful and timely issue in the current era. In addition, the study applied through statistical analysis to examine the preventive behaviors among different populations worldwide. The major strength also includes the theoretical background. Although the studied sample is not large if considering this is a worldwide recruitment, the study does have its merits and the logical flaw is smooth. Nevertheless, some improvements are needed for this work.

1. There are some papers using behavioral model to examine the preventive behaviors during the COVID-19 outbark period, the authors should cite these papers and have comparisons with these papers as they are relevant to this topic. Please see the following.

Chang, K.-C., Strong, C., Pakpour, A. H., Griffiths, M. D., & Lin, C.-Y. (2020). Factors related to preventive COVID-19 infection behaviors among people with mental illness. Journal of the Formosan Medical Association. https://doi.org/10.1016/j.jfma.2020.07.032

Lin, C.-Y., Imani, V., Majd, N. R., Ghasemi, Z., Griffiths, M. D., Hamilton, K., Hagger, M. S., & Pakpour, A. H. (2020). Using an Integrated Social Cognition Model to Predict COVID-19 Preventive Behaviours. British Journal of Health Psychology. https://doi.org/10.1111/bjhp.12465

Lin, C.-Y., Broström, A., Griffiths, M. D., & Pakpour, A. H. (2020). Investigating mediated effects of fear of COVID-19 and COVID-19 misunderstanding in the association between problematic social media use and distress/insomnia. Internet Interventions, 21, 100345. doi:10.1016/j.invent.2020.100345

Hagger, M. S., Smith, S. R., Keech, J. J., Moyers, S. A., & Hamilton, K. (2020). Predicting Social Distancing Intention and Behavior During the COVID-19 Pandemic: An Integrated Social Cognition Model. Annals of Behavioral Medicine. https://doi.org/10.1093/abm/kaaa073

2. The authors should use the following references to portray the importance and conditions of preventive behaviors during COVID-19 outbreak. Specifically, Rieger in his two papers describes the intention to perform some preventive behaviors among Germany. Shrivastava and Shrivastava mentioned the shortage of personal protective equipment during the COVID-19 outbreak period. Therefore, with proper implementation of preventive behaviors, the problem of shortage in personal protective equipment may be resoled (i.e., the population may not be panic on the outbreak and not crazily purchase the personal protective equipment). Lin and Cheng discussed the effectiveness of successful government policy. Therefore, the findings presented in the current study may take reference from Lin and Cheng to activate governments in doing some reactions timely and immediately.

Rieger MO. Triggering altruism increases the willingness to get vaccinated against COVID-19. Soc Health Behav 2020;3:78-82

Rieger MO. To wear or not to wear? Factors influencing wearing face masks in Germany during the COVID-19 pandemic. Soc Health Behav 2020;3:50-4

Shrivastava SR, Shrivastava PS. COVID-19 pandemic: Responding to the challenge of global shortage of personal protective equipment. Soc Health Behav 2020;3:70-1

Lin MW, Cheng Y. Policy actions to alleviate psychosocial impacts of COVID-19 pandemic: Experiences from Taiwan. Soc Health Behav 2020;3:72-3

3. I wondering why the authors did not use any assessments that have been designed for COVID-19. Specifically, Ahorsu et al. have developed the Fear of COVID-19 Scale; Lee has developed the Coronavirus Anxiety Scale and Obsession with COVID-19 Scale; Taylor et al. have developed the COVID Stress Scales; Chang et al. have developed Preventive COVID-19 Infection Behaviors Scale. The authors should at least acknowledge this as one of the limitations. That is, mentioning in their Limitation section that these are the available instruments to assess COVID-19 impacts with proper citations.

4. As the sample is consisted of different ethnicity groups, I wonder whether the authors have all different languages used in the present study. Or, the authors only use English for all the measures? This information should be clearly indicated and if the latter is the truth, this should be acknowledged as one of the limitations.

5. In Figure 1, it is unclear what “Demographic 1” and “Demographic 2” indicate.

6. Please report all the p-values instead of using symbols to indicate whether the p-value is over 0.05, smaller 0.05, smaller 0.01, or smaller 0.001. For those p-values smaller than 0.001, the authors can use “<0.001” to present.

7. In Table 4, please provide the SE of the unstandardized coefficient. Also, please provide the R square and adjusted R square for each step.

8. I am not sure whether the authors have adjusted their type 1 error in all the t-tests and ANOVAs. It seems that they did a lot of inferential testing and the chance of type 1 error inflation is high.

6. PLOS authors have the option to publish the peer review history of their article (what does this mean?). If published, this will include your full peer review and any attached files.

Reviewer #1: No

---

## [Author Response · Author response to Decision Letter 0]

14 Nov 2020

Our responses are in the letter to the Editor

---

## [Decision Letter · Decision Letter 1]

14 Dec 2020

Predicting health behavior in response to the coronavirus disease (COVID-19):

Worldwide survey results from early March 2020

PONE-D-20-25877R1

Dear Dr. Anaki,

We’re pleased to inform you that your manuscript has been judged scientifically suitable for publication and will be formally accepted for publication once it meets all outstanding technical requirements.

Kind regards,

Amir H. Pakpour, Ph.D.

Academic Editor

PLOS ONE

Additional Editor Comments (optional):

Reviewers' comments:

Reviewer's Responses to Questions

**Comments to the Author**

1. If the authors have adequately addressed your comments raised in a previous round of review and you feel that this manuscript is now acceptable for publication, you may indicate that here to bypass the “Comments to the Author” section, enter your conflict of interest statement in the “Confidential to Editor” section, and submit your "Accept" recommendation.

Reviewer #1: All comments have been addressed

2. Is the manuscript technically sound, and do the data support the conclusions?

Reviewer #1: Yes

3. Has the statistical analysis been performed appropriately and rigorously? 

Reviewer #1: Yes

4. Have the authors made all data underlying the findings in their manuscript fully available?

Reviewer #1: Yes

5. Is the manuscript presented in an intelligible fashion and written in standard English?

Reviewer #1: Yes

6. Review Comments to the Author

Reviewer #1: The authors have responded to all my prior comments and they are satisfactory. Therefore, I have no further comments on this work and would like to recommend publication.

7. PLOS authors have the option to publish the peer review history of their article (what does this mean?). If published, this will include your full peer review and any attached files.

Reviewer #1: No

---

## [Editor Report · Acceptance letter]

22 Dec 2020

PONE-D-20-25877R1 

Predicting health behavior in response to the coronavirus disease (COVID-19): Worldwide survey results from early March 2020 

Dear Dr. Anaki:

I'm pleased to inform you that your manuscript has been deemed suitable for publication in PLOS ONE. Congratulations! Your manuscript is now with our production department. 

Kind regards, 

on behalf of

Dr. Amir H. Pakpour 

Academic Editor

PLOS ONE